# Toxicological Assessments of a Pandemic COVID-19 Vaccine—Demonstrating the Suitability of a Platform Approach for mRNA Vaccines

**DOI:** 10.3390/vaccines11020417

**Published:** 2023-02-11

**Authors:** Cynthia M. Rohde, Claudia Lindemann, Michael Giovanelli, Rani S. Sellers, Jan Diekmann, Shambhunath Choudhary, Lila Ramaiah, Annette B. Vogel, Yana Chervona, Alexander Muik, Ugur Sahin

**Affiliations:** 1Drug Safety Research and Development, Pfizer Worldwide Research, Development & Medical, Pfizer, Inc., Pearl River, NY 10965, USA; 2BioNTech SE, 55131 Mainz, Germany; 3Drug Safety Research and Development, Pfizer Worldwide Research, Development & Medical, Pfizer, Inc., Groton, CT 06340, USA; 4Department of Pathology and Laboratory Medicine, University of North Carolina, Chapel Hill, NC 27599, USA

**Keywords:** BNT162b1, BNT162b2, BNT162b3, COVID-19 mRNA vaccine, rat, toxicity, nonclinical safety

## Abstract

The emergence of SARS-CoV-2 at the end of 2019 required the swift development of a vaccine to address the pandemic. Nonclinical GLP-compliant studies in Wistar Han rats were initiated to assess the local tolerance, systemic toxicity, and immune response to four mRNA vaccine candidates encoding immunogens derived from the spike (S) glycoprotein of SARS-CoV-2, encapsulated in lipid nanoparticles (LNPs). Vaccine candidates were administered intramuscularly once weekly for three doses at 30 and/or 100 µg followed by a 3-week recovery period. Clinical pathology findings included higher white blood cell counts and acute phase reactant concentrations, lower platelet and reticulocyte counts, and lower RBC parameters. Microscopically, there was increased cellularity (lymphocytes) in the lymph nodes and spleen, increased hematopoiesis in the bone marrow and spleen, acute inflammation and edema at the injection site, and minimal hepatocellular vacuolation. These findings were generally attributed to the anticipated immune and inflammatory responses to the vaccines, except for hepatocyte vacuolation, which was interpreted to reflect hepatocyte LNP lipid uptake, was similar between candidates and resolved or partially recovered at the end of the recovery phase. These studies demonstrated safety and tolerability in rats, supporting SARS-CoV-2 mRNA-LNP vaccine clinical development.

## 1. Introduction

The rapid development of vaccines against severe acute respiratory syndrome coronavirus 2 (SARS-CoV-2), the virus responsible for the ‘coronavirus disease 2019′ (COVID-19) pandemic, showcases the capability of modern molecular technologies for vaccine development. The first human clinical trials of a COVID-19 vaccine were initiated within 3–4 months of the publication of the SARS-CoV-2 sequence [1]. This rapid vaccine generation was the result of advances in ribonucleic acid (RNA) and lipid nanoparticle (LNP) technology allowing for the development of RNA-based vaccines [2,3]. The ease of development and manufacture of messenger RNA (mRNA) vaccines is improved over traditional recombinant protein-based vaccines and lacks the human health risks associated with manufacture of virus for inactivated or modified live viral vaccines [4,5]. Unlike live attenuated vaccines, RNA vaccines are not infectious and may be given to people who cannot be administered a live virus vaccine, such as pregnant women and immunocompromised individuals [6,7]. Further, RNA does not integrate into the host genome, and it is metabolized and eliminated by the body’s natural mechanisms [6,8]. Although there were no licensed vaccines using this modality at the beginning of the COVID-19 pandemic, mRNA-based technologies had been evaluated in an assortment of clinical trials for both prophylactic and therapeutic indications without evidence of safety concerns, and as of August 2022, two LNP-mRNA vaccines have been approved for licensure [9,10,11,12]. Therefore, the LNP-mRNA vaccines, as well as mRNA-based technologies, present a valid platform for future indications with a demonstrated potential for fast-track development.

mRNA-based vaccines are single-stranded, 5′-capped mRNAs that contain an inserted coding sequence for the target protein, which is translated by the host cell machinery after release into the cell cytoplasm [6]. RNA is a highly versatile molecule that can be manufactured using a cell-free in vitro transcription process; the molecule can be quickly modified to express almost any protein antigen of interest, making it highly adaptable for use in developing vaccines against emerging pathogens [6,13]. Modifications to the 5′ cap structure, 5′ and 3′ untranslated regions (UTRs), poly-A tail, and the coding sequence may increase both the mRNA half-life and efficiency of target protein translation [14,15]. The inclusion of nucleoside analogs, such as 1-methyl-pseudouridine in place of uridine, may also decrease targeted degradation and reduce RNA-mediated innate immune responses that may limit tolerability in humans [16,17].

Coronaviruses, such as SARS-CoV-2, are (+)ssRNA-enveloped viruses that encode for a total of four structural proteins [18]. Of these four structural proteins, the Spike glycoprotein (S protein) is the key target for vaccine development, as it mediates host-cell entry [19]. The receptor binding domain (RBD) of the S protein binds to the host cell receptor, angiotensin-converting enzyme 2 (ACE2) [20]. The S protein of SARS-CoV-2 mediates host cell entry in two ways: receptor-mediated endosomal host cell entry or viral membrane fusion with the host cell plasma membrane [21].

Induction of S protein-specific immune responses is key to virus neutralization. Four different LNP-formulated, nucleoside-modified mRNA (modRNA) vaccine candidates targeting the S protein were evaluated in nonclinical rat studies as part of the joint Pfizer-BioNTech COVID-19 RNA vaccine development program. Using the same mRNA backbone, two candidates encoded the RBD (BNT162b1 and BNT162b3), while the other two encoded the full-length S protein in its pre-fusion conformation (BNT162b2 [V8] and BNT162b2 [V9]). BNT162b2 (V9) is the clinically approved Pfizer-BioNTech mRNA vaccine against COVID-19. It is a codon-optimized version of BNT162b2 (V8), which encodes for identical amino acid sequences but with improved protein expression and immunogenicity.

The mRNA component of the BNT162b vaccine candidates is encapsulated into LNPs, which protect the RNA from degradation and enable transfection of the RNA into host cells after intramuscular (IM) injection [22]. The LNP used for the vaccine candidate was composed of four different lipids: cholesterol, an ionizable amino lipid, a PEGylated phospholipid, and a phosphocholine lipid. mRNA-lipid nanoparticles form upon the mixing of the RNA with the dissolved lipids, encapsulating the mRNA [23]. After injection, the mRNA-LNPs are taken up by host cells (e.g., muscle cells, tissue-resident, or recruited antigen-presenting cells), where the mRNA is released into the cytoplasm [6,13]. mRNA-LNPs can also directly reach immune cells in the draining lymph nodes via lymphatic vessels. Using the host cell machinery, the encoded S antigen is translated, and the expressed protein is presented at the cell surface, released, or presented by major histocompatibility complex (MHC) I or II (depending on the cell type) where it can initiate antigen-specific and protective humoral and cell-mediated immune responses [6,24]. The distribution of the mRNA-LNPs determines the mRNA expression pattern [25,26], with the majority of the LNP remaining at the injection site after IM injection. LNP components are metabolized and recycled or eliminated.

All four vaccine candidates were evaluated in two repeat dose toxicity studies in Wistar Han rats, which included a 3-week recovery phase. The objective of these studies was to evaluate the safety and immunogenicity of the vaccine candidates. This manuscript details the nonclinical study data for the four candidates and demonstrates the safety of this mRNA-LNP platform in the rat. 

## 2. Materials and Methods

### 2.1. Animals and Husbandry

All animal care and experimental procedures were conducted in compliance with guidelines for the care and use of laboratory animals [27] as well as local regulations. Each study was conducted according to GLP and OECD guidelines.

Male and female Wistar Han rats were supplied by Charles River Laboratories Germany GmbH, Sulzfeld, Germany (8–9 weeks at dosing start) for Study 1 or Charles River Laboratories, Raleigh, NC USA (9 weeks at dosing start) for Study 2. Rats were selected as the test species because they are commonly used in toxicity studies, have a large historical database, and produce an antigen-specific immune response to the S protein encoded by BNT162b mRNA.

Animals were offered food (certified commercial pellet diet, ssniff^®^ R/M-H V1534, ssniff Spezialdiäten GmbH, Soest, Germany [Study 1] or Certified Irradiated Rodent Diet 5002, PMI Feeds Inc., St Louis, MO, USA [Study 2]) and locally sourced water ad libitum, except when fasting was conducted prior to clinical pathology collections or euthanasia. Animals were housed individually throughout the study in MAKROLON cages (type III plus) with granulated textured wood (Granulat A2, J. Brandenburg, Goldenstedt, Germany) for Study 1 or in polycarbonate cages with ALPHA-dri^®^ (Shepherd Specialty Papers Inc.) for Study 2. Environmental conditions across studies were set to maintain relative humidity ranging from 30% to 70% and temperature ranging from 66°F to 79°F with room lighting set to provide a 12-h light/dark cycle.

### 2.2. Test and Control Articles

BNT162b1, BNT162b2 (V8 and V9), and BNT162b3 are LNP-formulated, modRNA vaccines for active immunization to prevent COVID-19 (Table 1). 

BNT162b1 encoded a secreted form of the RBD of the SARS-CoV-2 S protein. BNT162b2 encoded a prefusion stabilized, membrane-anchored SARS-CoV-2 full-length S protein. Two versions of BNT162b2, variants 8 (V8) and 9 (V9), were evaluated non-clinically. These two variants differed only in their codon optimization sequences, which were designed to improve antigen expression; the amino acid sequences of the encoded antigens were identical. BNT162b3 also encoded the RBD of the SARS-CoV-2 S protein; however, it was a membrane-bound form. 

For both studies, the test material was supplied by Polymun Scientific, Klosterneuburg, Austria. The RNA drug substances of BNT162b vaccines were highly purified single-stranded, 5′-capped mRNAs produced by in vitro transcription from corresponding DNA templates [28]. Each modRNA contained 1-methylpseudouridine instead of uridine. 

The LNP component of the vaccine was comprised of two functional lipids ALC-0315 ((4 hydroxybutyl)azanediyl)bis(hexane-6,1-diyl)bis(2-hexyldecanoate)) and PEGylated-ALC-0159 (2-[(polyethylene glycol)-2000]-N,N-ditetradecylacetamide) and two structural lipids DSPC (1,2-distearoyl-sn-glycero-3-phosphocholine) and cholesterol at fixed ratios. 

The test material was a frozen solution for injection at a concentration of approximately 0.5 mg formulated RNA/mL. The test material was thawed prior to administration between 2 °C and 25 °C.

The control article used in Study 1 was phosphate-buffered saline/300 mM sucrose (PBS/Sucrose; Polymun Scientific Immunbiologische Forschung GmbH, Klosterneuburg, Austria). In Study 2, the control used was 0.9% sterile saline for injection, USP (B Braun Medical Inc, Bethlehem, PA, USA).

### 2.3. Study Design

Two repeat-dose toxicity studies were conducted to support the development of the Pfizer-BioNTech COVID-19 vaccine candidates (BNT162b1, BNT162b2 [V8], BNT162b2 [V9], and BNT162b3). The first study was conducted at the Laboratory of Pharmacology and Toxicology GmbH & Co. KG, Hamburg, Germany (Study 1). The second study was conducted at Pfizer, Groton, CT, USA (Study 2). Both studies were similar in design, as outlined in Figure 1.

Briefly, male and female (15/sex/group) Wistar Han rats were acclimated, randomly assigned to groups, and then administered once weekly IM injections of the control material or 1 of the 4 BNT162b modRNA vaccine candidates for a total of 3 doses. IM administration was selected as this was the clinical route of vaccine administration. A subset of animals (10/sex/group) was euthanized by carbon dioxide or isoflurane inhalation followed by exsanguination 2 days after the third dose (Day 17), while the remaining animals (5/sex/group) underwent an approximate 3-week recovery and then were euthanized (~Day 38). In Study 1, rats were administered either PBS/Sucrose, 30 or 100 µg RNA/dosing day BNT162b1, or 100 µg RNA/dosing day BNT162b2 (V8). An additional 3 animals/sex/groups were included in this study for cytokine sample analysis. These satellite animals were supplied with a pre-implanted femoral vein catheter for repeated blood sampling (Charles River Laboratories Germany GmbH, Sulzfeld, Germany). In Study 2, rats were administered saline, 30 µg RNA/dosing day BNT162b2 (V9), or 30 µg RNA/dosing day BNT162b3. In Study 1, the dose of 100 µg was included as it was the highest intended clinical dose to be evaluated in Phase 1 clinical trial. The 30 µg dose reflected the selected dose for Ph 2/3 clinical development. IM injections were administered into the right and/or left biceps femoris in Study 1 or the left quadriceps muscle of the hindlimb in Study 2. The dose volumes used in both studies are indicated in Appendix A.

### 2.4. In Life Assessments

Body weights were recorded at least once prior to dose initiation and at least twice weekly during the dosing and recovery phases, including predose on each dosing day. A fasted body weight was recorded prior to each necropsy. Quantitative food consumption was recorded once or twice weekly.

Clinical observations occurred at least once daily prior to the initiation of dosing, at least twice daily on non-dosing days and during the recovery phase, and prior to and after each dose on dosing days. Body temperatures (rectal) were recorded 4 and 24 h after each dose in both studies, as well as once prior to the initiation of dosing and predose on dosing days in Study 2 and weekly during the recovery phase in Study 1. If an animal had a body temperature ≥ 40 °C (predose body temperature upper limit from internal historical data) at 24 h postdose, additional monitoring was conducted. 

Local reactions at the injection site were assessed using the Draize scoring method in both studies [29]. In Study 2, reactions were assessed predose as well as 4 and 24 h postdose (Appendix A), and if an animal had irritation present (e.g., severity score of ≥2) at 24 h postdose, additional monitoring was conducted. In Study 1, reactions were assessed at 4, 24, and 48 h postdose; if irritation was present at the 48-h postdose observation, additional monitoring was conducted.

In Study 1, ophthalmoscopic and auditory evaluations were conducted in all study animals prior to dose administration and near the end of the dosing phase (Day 16), and in the recovery phase, animals near the end of recovery (Day 37). In Study 2, ophthalmoscopic examinations were conducted on all animals prior to dose administration and near the end of the dosing phase (Day 15 [male] or 16 [female]). As there were no findings at the end of the dosing phase, examinations were not conducted at the end of the recovery phase.

### 2.5. Blood and Urine Sample Collections for Clinical Pathology

Samples for clinical pathology (hematology, coagulation, clinical chemistry, acute phase proteins, and urinalysis) were collected on Days 4 (nonterminal; hematology, clinical chemistry, and acute phase proteins only), 17 (terminal), and 38 or 39 (terminal), from overnight fasted animals. For non-terminal collections, hematology was assessed in the first 5 or 7 animals/sex/group, and clinical chemistry was assessed in the last 5 or 8 animals/sex/group (Study 1 or 2, respectively). Phlebotomy sites used were the retrobulbar venous plexus (under isoflurane anesthesia) for Study 1 or the jugular vein (non-terminal, under isoflurane anesthesia) or aorta under isoflurane anesthesia followed by exsanguination (terminal) for Study 2. Blood samples were collected into appropriate tubes (K2EDTA for hematology, 3.2% sodium citrate for coagulation, lithium heparin or serum separator for clinical chemistry, and acute phase proteins [for Study 1 or 2, respectively]).

### 2.6. Hematology, Coagulation, Clinical Chemistry, Urinalysis, and Acute Phase Protein Analysis

The specific parameters assessed can be found in Appendix A. Hematology was evaluated using a Siemens Advia 120 analyzer (Siemens Healthineers, Erlangen, Germany) for Study 1 and a Siemens Advia 2120i analyzer (Siemens Healthineers Tarrytown, NY, USA) for Study 2. Fibrinogen activated partial thromboplastin time, and prothrombin time was evaluated on the Amax Destiny Plus (Tcoag Deutschland 6mbh 32657, Lemgo, Germany) for Study 1 and on the Diagnostic Stago STA-R evaluation coagulation analyzer (Diagnostic Stago, Parsippany, NJ, USA) for Study 2. For Study 2, blood smears were prepared for the first 7 animals on Day 4 and all animals on Day 17 and Day 38. Blood cell morphology was evaluated microscopically on 5 animals of each sex from all groups at both scheduled necropsies (i.e., at dosing and recovery phases). Routine clinical chemistry parameters were evaluated using a Konelab 30i (Thermo Fisher Scientific 63303, Drieieich, Germany) for Study 1 and a Siemens Advia 1800 clinical chemistry analyzer (Siemens Healthineers, Tarrytown, NY, USA) for Study 2. For Study 1, acute phase proteins alpha-2 macroglobulin (A2M) and alpha-1-acid glycoprotein (A1AGP) were measured by ELISA (Rat Alpha 1 Acid Glycoprotein/AGP ELISA Kit, ab157729); Rat alpha 2 Macroglobulin ELISA Kit, ab157730). For Study 2, acute phase proteins were measured using the rat MSD Acute Phase Protein Panel 1 on the MSD SECTOR S 600 Analyzer (Meso Scale Design). Routine urinalysis parameters were measured. For Study 2, a microscopic examination of sediment for formed elements was performed on 5 animals of each sex from all dose groups at both scheduled necropsies (i.e., dosing and recovery phases). 

### 2.7. Cytokine Analysis

Cytokines (IFN-γ, TNF-α, IL-1β, IL-6, and IL-10) were evaluated in Study 1. Blood samples for cytokine analysis were collected from the femoral vein catheter of satellite animals on Days 1, 8, and 15 (prior to and 6 h following each dose) and on Day 17 (48 h following the 3rd dose). Two 75 µL aliquots of serum per animal per timepoint were stored at −20 °C ± 10% prior to analysis by cytometric bead array (ProcartaPlex) using a Cytomics FC 500 flow cytometer (Beckman Coulter GmbH, 47704 Krefeld, Germany).

### 2.8. Serology 

#### 2.8.1. Sample Collection and Processing

In Study 1, blood was collected by retro-orbital sampling under isoflurane anesthesia from all study animals at the end of the dosing phase (Day 17) and in all recovery animals at the end of the recovery phase (Day 38). In Study 2, blood was collected into serum separator tubes under anesthesia from the jugular vein (nonterminal) from all study animals on Day 8 prior to initiation of dosing (Day 5) and under anesthesia from the aorta (terminal) at the end of the dosing phase (Day 17) and in all recovery animals at the end of the recovery phase (Day 38). Samples were processed into serum and stored at −60 °C or lower until analysis.

#### 2.8.2. Analysis of S1- and RBD-Specific Serum IgG 

For Study 1, recombinant SARS-CoV-2 S1 protein (1 µg/mL, Sino Biological, Eschborn, Germany) was coated overnight on MaxiSorp plates (Thermo Fisher Scientific, Drieieich, Germany) using sodium carbonate buffer. After a blocking step the next day, serum samples were incubated for 1 h at 37 °C to allow the binding to antigens. Bound rat IgG was detected using a horseradish peroxidase (HRP)-conjugated secondary antibody (Jackson ImmunoResearch, West Grove, PA, USA) and tetramethylbenzidine (TMB) substrate (Biotrend, Köln, Germany). Data collection was performed using a BioTek Epoch reader and Gen5 software version 3.0.9. Each serum sample dilution was tested in duplicates. In parallel, a serial dilution of a polyclonal IgG rat isotype control (Southern Biotech, Birmingham, AL, USA) was implemented, allowing for concentration calculation as the sample signals were correlated to a standard curve of the isotype control.

#### 2.8.3. VSV-SARS-CoV-2 S Pseudovirus Neutralization Assay

For Study 1, a pseudovirus neutralization assay was used to determine the neutralizing antibody titers. The method described previously [28] also included the correlation verification to a SARS-CoV-2-based virus neutralization assay. In brief, a recombinant and replication-deficient vesicular stomatitis virus (VSV) encoding the green fluorescent protein but not the VSV glycoprotein (VSVΔG-GFP) was pseudotyped with SARS-CoV-2 S protein. For the pseudovirus neutralization assay (pVNT), serial dilutions of the rat serum samples from Study 1 were prepared and pre-incubated for 10 min at room temperature with VSV/SARS-CoV-2 pseudovirus suspension. After the following incubation on Vero-76 cells for 20 h at 37 °C, plates were analyzed using the IncuCyte Live Cell Analysis system (Sartorius, IncuCyte 2019B Rev2 software). The 90% pseudovirus neutralization titer (pVNT_90_) was reported as the reciprocal of the highest dilution of serum, still yielding a 90% reduction in GFP-positive infected cell number per well compared to the mean of the no serum pseudovirus positive control. Each serum sample dilution was tested in duplicates.

#### 2.8.4. SARS-CoV-2 Neutralization Assay

For Study 2, a SARS-CoV-2 neutralization assay was used to assess the antigen-specific functional immune response. Each serum sample was tested in duplicate for serological detection of SARS-CoV-2 specific neutralizing antibodies by VisMederi (Siena, Italy). The cytopathic effect (CPE) based SARS-CoV-2 microneutralization assay is a 4–5 day manual 96-well assay. Vero E6 cells (CRL-1586, American Type Culture Collection) were seeded into 96-well tissue culture plates. Serial dilutions of test sera were incubated with SARS-CoV-2 2019 nCOV ITALY/INMI1 infectious virus to allow any antigen-specific antibodies to bind to the virus. The serum-virus mixture was then transferred onto the Vero cell monolayer and allowed to incubate for 3–4 days to allow for infection by the non-neutralized virus to occur. Plates were visualized under an inverted light microscope, and a sample microneutralization titer (VNT) was determined. The VNT was defined as the reciprocal of the highest serum dilution that protects at least 50% of the cells from CPE. If no neutralization was observed (VNT50 < 10), an arbitrary titer value (5) of half of the limit of detection [LOD] was reported. The LOD is defined by the applied starting dilution.

#### 2.8.5. Post-Mortem Assessments

Complete necropsies, tissue collection, organ weights, and macroscopic tissue evaluation were performed on all animals. Necropsy includes a macroscopic examination of the external surface of the body, the thoracic and abdominal cavities and their contents, and the collection of all major tissues and macroscopic findings (Appendix A).

Selected organs from all animals were weighed at the scheduled necropsy (Appendix A). Organ-to-body weight and organ-to-brain weight ratios in Study 2 and organ-to-body weight ratios in Study 1 were calculated.

Representative samples of collected tissues were fixed in 10% (Study 2) or 7% (Study 1) neutral buffered formalin except for the eye with optic nerve attached (Davidson’s) and testis and epididymis (modified Davidson’s). All tissues were processed for slide preparation and stained with hematoxylin and eosin.

For the dosing phase, all tissues (excluding the larynx) collected from all dosing phase animals were examined microscopically. In Study 1, all tissues examined at the end of the recovery phase were identical to those evaluated at the end of the dosing phase. In Study 2, microscopic evaluation of recovery phase tissues in all animals was limited to real or anticipated target organs: bone marrow (sternum), joint, liver, draining lymph node, inguinal lymph node, macroscopic findings, skeletal muscle, injection site, and spleen. Microscopic findings were graded on a scale of 1 to 5 as minimal, mild, moderate, marked, or severe; findings not graded were listed as present. The type of infiltrating cells in tissues was based on the morphology of their nucleus, their size, the appearance of cytoplasm, and in the case of granulocytes, how their granules stain.

#### 2.8.6. Statistical Analysis

In Study 1, bodyweight, body weight change, food consumption, hematology and coagulation, clinical chemistry, acute phase proteins, cytokines, urinalysis, relative and absolute organ weights were analyzed using Bartlett’s test to assess the equality of variance across groups, and Shapiro-Wilk’s test was used to assess the normality of the data distribution in each group. Data with homogeneous variances and normal distribution in all groups were analyzed using ANOVA followed by Dunnett’s test. In case of heterogeneity and/or nonnormality of distribution, the stepwise transformation of the values into logarithmic or rank values was performed prior to ANOVA. Statistical tests were conducted at the 5% and 1% significance levels.

In Study 2, statistical analysis of body weight, body weight change, food consumption, body temperature, hematology and coagulation, clinical chemistry, acute phase proteins, urinalysis, relative and absolute organ weights, and injection site reactions were conducted. Descriptive statistics were generated for each parameter and group at each scheduled sampling time or each time interval. Statistical tests were conducted at the 5% and 1% significance levels. Analysis of body temperature was based on the maximum body temperature post-injection for each animal. Analysis of injection site score was based on the average irritation score post-injection for each animal. A nonparametric (rank-transform) one-way analysis of variance (ANOVA) was conducted, with 2-sided pairwise comparison of each dose group to the reference group (saline control) using Dunnett’s test. Average ranks were assigned to ties.

## 3. Results

### 3.1. In-Life Findings

In both repeat-dose toxicity studies, administration of BNT162b1, BNT162b2, or BNT162b3 was tolerated without evidence of systemic toxicity at all doses evaluated. There were no vaccine-related mortalities, clinical signs, or effects on ophthalmic or auditory parameters. In Study 1, mean body weights were lower 24 h after each BNT162b1 and BNT162b2 (V8) vaccine administration compared with control animals (0.91x to 0.95x BNT162b1 and 0.89x BNT162b2 [V8]). Body weight gain between the vaccine administrations was comparable to the control group. There were no noteworthy effects on body weight at the end of the recovery phase (Appendix A). In Study 2, there were lower mean body weights (0.93x−0.94x control) in males administered 30 µg BNT162b3 on Days 11 and 15 compared with saline controls. Test article-related higher mean body weight (1.05x−1.06x control) was noted in males only during the recovery phase on Days 28, 32, 35, and 38 for animals administered 30 µg BNT162b2 (V9). No test article-related body weight changes were noted for animals administered BNT162b3 during the recovery phase (Appendix A).

BNT162b1 and BNT162b2 (V8)-administered animals generally had higher body temperatures compared with control animals at 4 and 24 h postdose. Group means temperatures in rats administered the 100 µg BNT162b1 and BNT162b2 (V8) vaccine were higher but within approximately 1 °C of the group mean body temperature of control animals. Animals in the BNT162b2 (V9) and BNT162b3 groups also had higher body temperatures compared with control animals on Day 1 (up to 0.54 °C and 0.71 °C, respectively), Day 8 (up to 0.98 °C and 1.26 °C, respectively), and Day 15 (up to 1.03 °C and 1.09 °C, respectively). No animals administered BNT162b1, BNT162b2, or BNT162b3 vaccines had body temperatures > 40.0 °C after administration.

Local reactions were observed in male and female animals dosed IM with BNT162b1, BNT162b2 (V8 and V9), and BNT162b3. The incidence and severity of the reactions were higher after the second or third administrations compared with the first administration. In Study 1, the majority of the animals had very slight edema or rarely slight erythema after the first dose. After the second (48 h postdose) or third dose (24 h postdose), the severity of edema increased up to moderate or rarely, severe grades (BNT162b2 [V8]) in some animals from all vaccine-administered groups (Figure 2).

An increase in severity grade for edema and erythema was also dose-dependent, with higher severity noted with the administration of 100 µg BNT162b1 as compared with 30 µg BNT162b1. Increased erythema was present in females administered 100 µg BNT162b1 and in females and males administered 100 µg BNT162b2 (V8) 144 h after the second dose (Day 14) (Figure 3).

The edema and erythema resolved prior to the next injection or, in recovery animals, during the 3-week recovery phase. Similarly, in Study 2, the majority of the animals had very slight edema or slight erythema after the first dose. The incidence and severity of the edema were higher in the BNT162b2 (V9) and BNT162b3 after the second or third administrations compared with the first administration (Figure 2). Injection site erythema (Figure 3) and edema were fully resolved prior to dose administration on Days 8 and 15 and by the end of the 3-week recovery phase (Figure 3).

### 3.2. Clinical Pathology and Acute Phase Proteins

In both repeat-dose toxicity studies, the administration of BNT162b21, BNT162b2 (V8), BNT162b2 (V9), and BNT162b3 resulted in clinical pathology changes that were consistent with anticipated immune responses to the vaccine, including an inflammatory leukogram and an acute phase response (Figure 4 and Appendix A).

Hematology findings consistent with an inflammatory leukogram included elevations in white blood cells, including neutrophils, monocytes, eosinophils, basophils, and large unstained cells (LUC). Findings consistent with an acute phase response included elevations in positive acute phase reactants (alpha-1-acid glycoprotein, alpha-2 macroglobulin, fibrinogen, and globulins) and reductions in albumin, a negative acute phase reactant, and associated reductions in albumin:globulin (A:G) ratio. Hypersegmented neutrophils were present on peripheral blood smears of test article-treated animals. These findings were generally greatest on Day 17, 48 h after the last injection. Other findings that were inconsistently seen in one or both studies and were considered to be secondary to inflammation included slight reductions in platelet counts on Day 17, transient reductions in reticulocyte counts on Day 4 followed by elevations on Day 17, slight reductions in red blood cell mass (hemoglobin, red blood cells, and hematocrit) on Days 4 and/or 17, and alterations in mean cell hemoglobin (MCH), mean cell hemoglobin concentration (MCHC) and red cell distribution width (RDW) that were secondary to changes in reticulocytes [30,31,32] (Figure 4). The magnitude of many BNT162b1- or BNT162b2-related changes was dose-related (A2M, A1AGP, albumin, AG ratio, globulins, and most leukocytes) or only seen at 100 μg (platelets). Finally, there were elevations in γ-glutamyl transferase (GGT) in Study 1 that were not replicated in Study 2. All clinical pathology changes recovered by the end of the recovery phase (Day 38) except for elevations in RDW and globulins and reductions in A:G ratios.

### 3.3. Cytokines

There were no vaccine-related cytokine changes (IFN-gamma, TNF-α, IL-1β, IL-6, and IL-10) in Study 1 on Days 1, 8, or 15 (prior to and 6 h following each dose) or on Day 17 (48 h following the 3rd dose).

### 3.4. Organ Weight/Macroscopic and Microscopic Observations

Organ weight changes, macroscopic postmortem findings, and microscopic histopathological findings were generally comparable in both repeat-dose toxicity studies with the administration of BNT162b1, BNT162b2 (V8 or V9), or BNT162b3 vaccines. 

Macroscopic findings associated with the vaccines were variably present and included discoloration of the injection site (dark or pale), thickened/firm or enlarged injection site, enlarged draining and inguinal lymph nodes, and/or enlarged spleen. At the end of the 3-week recovery phase, the macroscopic injection site and spleen findings had fully recovered, and there was partial recovery of the enlarged draining and inguinal lymph nodes.

In both studies, vaccine administration resulted in an increase in absolute and relative (to body and/or brain weight) spleen weights in both sexes. Higher group mean absolute and relative spleen weights were noted in vaccine-administered males (1.14x–1.52x), and females (1.24x–1.62x) relative to the saline control group mean. There were no significant weight changes in other organs. Macroscopically, increased spleen weights correlated with the occasional observation of an enlarged spleen. Microscopically, spleen weight increases correlated with increased cellularity of hematopoietic cells in the red pulp and expansion of germinal centers (increased cellularity) in the white pulp. At the end of the 3-week recovery phase, the increased cellularity of hematopoietic cells in the red pulp of the spleen had fully recovered, whereas there was partial recovery of the increased cellularity of germinal centers in the white pulp. 

Microscopic findings at the injection sites included mixed cell inflammation and edema (Figure 5), which correlated with the macroscopic injection site findings. Inflammation at the injection site was characterized by large numbers of neutrophils with fewer plasma cells, macrophages, and lymphocytes admixed with abundant edema and small amounts of cellular debris, fibrin, and hemorrhage. Occasionally, inflammatory cells extend into the subcutaneous tissue/dermis of the overlying skin and into the extra-capsular tissue of the joint. 

At the end of the 3-week recovery phase, there was complete resolution of the edema and partial recovery of the inflammation; the inflammatory cell infiltrate more chronic in nature, comprised primarily of lymphocytes, plasma cells, and macrophages, with few neutrophils (Figure 6).

Increased cellularity was evident in the bone marrow, which was characterized by increased numbers of myeloid cells. This finding was fully resolved at the end of the 3-week recovery phase. The finding correlated with increased WBC in vaccinated animals.

Minimal hepatocellular vacuolation was evident in portal hepatocytes (Figure 7). This was not associated with hepatocyte degeneration or necrosis or with biochemical alterations in hepatocyte injury biomarkers (AST and ALT) and was considered secondary to the distribution of LNP lipids in the hepatocytes [33]. This finding was fully resolved at the end of the 3-week recovery phase.

Microscopic findings in enlarged draining and inguinal lymph nodes correlated with increased germinal center cellularity, similar to the spleen, as well as increased numbers of immature plasma cells (plasmablasts) (Figure 8). At the end of the 3-week recovery phase, the increased cellularity of germinal centers was partially recovered, and the immature plasma cells were replaced by mature plasma cells.

### 3.5. Serology

In order to gain insight into the immunogenicity of the different BNT162b candidates in rats, serum samples from animals were analyzed for antigen-specific immune responses. Two different kinds of assays were performed, analyzing the total antigen-specific IgG response as well as the neutralizing efficacy of the generated antibodies (Figure 9).

In Study 1, serum samples from two different time points, namely 17 days after the first administration as well as at the end of the study at Day 38, were used for analysis. All candidates at the given doses elicited a significant induction of IgG against S1 compared to the control group for both days of analysis, while between groups, no significant difference was detected. Seventeen days after the first immunization, the serum titers of IgG against S1 reached comparable geometric means (GM) between 100 µg doses of BNT162b1 (GM = 1579 µg/mL) and BNT162b2 (V8) (GM = 1589 µg/mL) followed by the 30 µg dose of BNT162b1 (GM = 1076 µg/mL). In parallel, samples were tested in a pVNT, and due to the high pVN titers of samples, the 50% pseudovirus neutralizing titers (pVNT_50_) were not defined as sample groups reached the upper limit of detection in the test assay. Therefore, 90% pseudovirus neutralizing titers (pVNT_90_) are reported, and animals that received 100 µg BNT162b2 (V8) showed the highest pVNT_90_ with a geometric mean of 192 followed by the 100 µg BNT162b1 group (GM of 146). Thirty-eight days after the first injection of the BNT162b candidates, overall titers increased, and 100 µg BNT162 induced both the highest IgG antibody response (GM for S1: 3086 µg/mL) as well as pVNT_90_ (GM of 946), including several samples reaching the upper limit of the quantification for the latter. Of note, for one animal in the 100 µg BNT162b2, no neutralization response was detected.

In Study 2, only neutralizing antibody titers were evaluated. Samples were collected prior to dosing, at the dosing phase necropsy (Day 17), and at the end of the 3-week recovery phase (Day 38). After immunization, BNT162b2 (V9) and BNT162b3 elicited SARS-CoV-2 neutralizing antibody responses in male and female rats at the end of the dosing and recovery phases of the repeat-dose toxicity study. On Day 17, group geometric mean 50% SARS-CoV-2 neutralizing titers (VNT_50_) were 1114 for males and 2501 for females administered 30 µg BNT162b2 (V9) and 993 for males and 1810 for females administered BNT162b3 compared with saline controls. By Day 38, group geometric mean VNT_50_ had risen to 5120 for animals (M and F) administered BNT162b2 (V9) and 3880 for animals (M and F) administered BNT162b3. SARS-CoV-2 neutralizing antibody responses were not observed in animals prior to vaccine administration or in saline-administered control animals.

## 4. Discussion

The nonclinical safety evaluation conducted for the Pfizer-BioNTech COVID-19 vaccine program demonstrated that all four modRNA BNT162b vaccine candidates were tolerated in rats when administered IM once weekly (3 total doses) at doses of 30 and/or 100 µg. Vaccine-related findings identified with all four modRNA vaccine candidates were similar, and some parameter changes (e.g., WBC, acute phase proteins, injection site scoring, S1-binding, and virus neutralizing titers) were slightly greater in animals administered 100 µg compared to those administered 30 µg, suggesting a dose-related difference in immune responses. Findings were attributed to immune activation, except for minimal microscopic periportal hepatocyte vacuolation, which was interpreted to reflect LNP lipid distribution to the liver [25]. This change was not associated with evidence of liver dysfunction or damage [26,34]. Findings identified at the end of dosing were generally resolved at the end of the recovery phase. For most study parameters, changes were typical of those identified in nonclinical vaccine studies including at the end of the recovery phase (e.g., partial recovery of enlarged lymph nodes, higher serum globulins, partial recovery of inflammation at the injection site) [35]. While the in-life and post-mortem findings were similar to other vaccine formulations tested non-clinically, the magnitude of changes, particularly in hematology and acute phase protein parameters, were somewhat higher than in non-mRNA-LNP vaccines [36]. This difference is interpreted to be the result of a strong innate immune activation by the mRNA-LNP platform. 

Recognition of the foreign antigen(s) that are part of any vaccine can lead to local and/or systemic reactogenicity. This can take the form of swelling, redness, or pain at the injection site, or if inflammatory mediators enter circulation, more systemic symptoms such as fever, fatigue, or headache [37]. Evidence of both types of reactions was observed in the rats after administration of the BNT162b vaccine candidates.

Injection site edema and erythema increased in severity after each vaccine administration. This may have been the result of the mRNA-LNP modality, but the weekly dosing interval might have also contributed to this observation. Routine nonclinical vaccine toxicity studies typically have vaccine administration at 2–3 week intervals; in an effort to accelerate vaccine safety studies, the dosing schedule was shortened. The 1-week interval between dose administration may have contributed to the already existing active innate immune responses at the injection site. Injection site reactogenicity was slightly greater in animals after the second or third dose of administered vaccines; similar trends in reactogenicity related to dose were identified in the human Phase 1 clinical trial [38]. However, in Phase 3 clinical trials, injection site reactogenicity did not increase after the second dose [39]. Microscopically, inflammation at the injection sites with BNT162b vaccine administration was within the range that has been reported in nonclinical vaccine studies [35,36,40]. At the end of the recovery phase, edema was no longer present at the injection site, and the inflammation was limited to scattered mononuclear cells (resolution phase) without fibrosis.

Nonadverse, small (<1.3 °C) increases in body temperature were transiently observed after each dose administration in rats administered the vaccine candidates compared with controls, suggesting similar increases in body temperature may be seen in humans. This was confirmed in Phase 3 clinical trial, with generally mild to moderate fever (body temperature ≥ 38 °C) reported in up to 16% of BNT162b2 recipients, mainly after the second dose [39].

Innate immune responses are essential in the development of strong and long-term vaccine efficacy, as they contribute to the development of antigen-specific T-cell and neutralizing antibody-generating plasma B-cell responses [41,42]. Immune responses in the draining lymph nodes and spleen led to the macroscopic observations of enlarged regional lymph nodes and, occasionally, enlarged spleens. This correlated with the expansion of germinal centers, consistent with B-cell activation, proliferation, and maturation, which are desired and expected responses to vaccine administration [43,44]. Microscopically, draining lymph nodes also had a notable expansion of immature plasma cells (plasmablasts) in the sinuses at the end of the dosing phase, which was replaced by mature plasma cells at the end of the recovery phase. This robust plasmablast expansion was more prominent than observed with other vaccine modalities tested by Pfizer and may underlie the robust antibody responses identified in nonclinical and clinical studies [38,45]. 

Innate immune responses also manifested as increased circulating white blood cells (particularly neutrophils), increased myelopoiesis in the bone marrow, and increased extramedullary hematopoiesis in the spleen. Such findings are common in nonclinical vaccine toxicity studies [35]. Microscopically, increased WBCs correlated with slight increases in the hematopoietic cells in the bone marrow (mostly myeloid cells) and, to a lesser extent, in the spleen at the end of the dosing phase. Inflammatory cytokines have been shown to play a critical role in the expansion of myeloid progenitors during inflammation-induced hematopoiesis [46,47]. Rodents have the capacity to generate myeloid and erythroid cells in the red pulp of the spleen with strong inflammatory signals, and thus this is an anticipated effect of immune stimulation in this species [48]. Somewhat unique to the mRNA-LNP vaccines compared to other Pfizer vaccine programs, however, was the presence of hypersegmented neutrophils both in the tissue section and on blood smears which fully recovered 3 weeks after the last dose administration. Hypersegmented neutrophil presence was considered to be secondary to the robust increases in neutrophil counts, likely related to the mobilization of bone marrow storage neutrophils, as well as prolonged neutrophil lifespan in circulation [49]. 

One unexpected hematology finding was the transiently lower reticulocyte counts on Day 4, which had recovered by the end of dosing on Day 17. This change had a slight and transient effect on the red blood cell parameters, which was not considered biologically impactful. The reduction in reticulocytes was likely the result of transitory suppression of erythropoiesis. A similar phenomenon has been reported 8–24 h after immune activation with an acute phase response [30,32,50]. The mechanism of reticulocyte effects has been suggested to be due to the sequestration of iron via TLR-mediated downregulation, degradation, and endocytosis of the iron exporter, ferroportin, limiting iron availability for hemoglobin synthesis [37,41,42]. An explanation for the lack of apparent reticulocyte decreases compared to control animals after the 3^rd^ vaccine administration may be due to the combined effect of a robust reticulocyte regenerative response and the age-related decrease in reticulocyte counts, as evidenced in control rats [51,52,53].In male and female control rats on Day 4 (Study 2), the mean reticulocyte counts were 392.1 and 301.7 × 10^3^/µL, respectively, which decreased by ~44–54% on Day 17 to 178.8 and 168.9 × 10^3^/µL, respectively. Interestingly, reticulocyte decreases were similarly observed in rats treated with the licensed LNP-siRNA pharmaceutical Onpattro™ but have not been observed in humans treated with this biotherapeutic [54], suggesting this may be a species-specific effect. 

In nonclinical vaccine toxicity studies, very slight decreases in platelets may be observed. Slightly lowered platelet counts (0.75x–0.66x concurrent controls) were identified on Day 17 in both sexes administered BNT162b vaccine candidates at 100 µg but not in those administered 30 µg doses, suggesting a dose-related impact on platelet counts. Regardless, the platelet counts in those groups were generally comparable to historical reference intervals, fully recovered 3 weeks after the last dose, and were not associated with coagulation time changes, evidence of bleeding, or microscopic changes in bone marrow megakaryocytes. The pattern of the changes in platelets was not consistent with autoimmunities such as the small magnitude of changes, transient nature of change, and high incidence. 

Consistent with the contention of a strong immune response to the BNT162b vaccine candidates, the magnitude of the acute phase response in the dosing phase, as measured by increases in the positive acute phase proteins alpha-1 acid glycoprotein (A1AGP), alpha-2-macroglobulin (A2M), and fibrinogen, was above that generally identified for aluminum-containing vaccines [36]. All measured acute phase markers had returned to control levels after a 3-week recovery period, except for globulins, which remained higher. 

The robust innate immune response to the BNT162b vaccine candidates suggested by the nonclinical data is likely the result of the mRNA-LNP platform. Single-stranded RNA (ssRNA) will activate innate immune responses via pathogen recognition receptors (PRRs) [34]. These include the endosomal receptors toll-like receptor (TLR)7 and TLR8, as well as the cytoplasmic RNA sensors including RIG-like receptors (RLRs) such as RIG-I, and NOD-like receptors (NLRs) such as NLRC2 [34,55]. These receptors upregulate pro-inflammatory cytokines, particularly type 1 interferons, which contribute to potent anti-viral responses. Modification of some or all the uridines by nucleoside analogs, such as 1-methylpseudouridine, as was the case with the BNT162b vaccine candidates, serves to dampen the innate immune response. 

Lipids may also have inherent adjuvant properties and can incite innate immune responses [56]. The lipid ALC-0315 is an ionizable lipid, which is a key component of the LNP and important for the delivery of nucleic acids. The ability of the LNP to encapsulate the RNA depends in large part on the ionizable nature of the lipid [57,58]. At a physiological pH, the ionizable lipid has a limited positive charge, which reduces clearance by the reticuloendothelial system [59,60], improving the LNP half-life. Unlike cationic lipids, the ionizable lipid has improved tolerability, as cationic lipids have been associated with toxicity and increased immune activation [61]. It has been demonstrated that the structure of the ionizable lipid in LNPs has an impact on the local reactogenicity and cytokine expression with parenteral administration [33,62,63]. Recent studies demonstrated that intradermal (ID) or IM administration of 10 μg of empty LNPs resulted in robust neutrophilic inflammation at the injection site, and LNP- encapsulating a polycystine RNA can activate multiple inflammatory pathways and induce IL-1β and IL-6 [62]. Injection site inflammation was comparable to saline control administration when the ionizable lipid was removed from the LNP, consistent with the reports of others that the ionizable lipid contributes to injection site reactogenicity after modRNA-LNP administration. Presently, it is unclear how the ionizable lipid components activate inflammation. However, the mechanisms may resemble those for the cationic lipids [64,65,66]. 

The lack of notable vaccine-related increases in IFN-γ, TNF-α, IL-1β, IL-6, and IL-10 at 6- and 48-h post-dose compared to controls was somewhat surprising. It is unclear why differences in cytokine concentrations between vaccinated and buffer control-administered were not readily evident, although there was wide inter-animal variability in cytokine concentrations which may have hindered interpretation. 

Higher GGT activities were identified in Study 1 (but not Study 2) and were interpreted to be the result of pre-analytical/analytical variables and not due to tissue damage. GGT is bile canalicular-facing membrane-bound enzyme expressed on bile duct epithelial cells (with a minimal expression on canalicular membranes of hepatocytes) [67,68,69,70]. Neither Study 1 nor Study 2 had biochemical evidence of hepatobiliary injury (such as higher ALT, ALP, or total bilirubin) or microscopic biliary changes. Differences that may have contributed to higher GGT in Study 1 but not Study 2 included the use of serum collection tubes with Li-heparin in Study 1 vs. without anticoagulant in Study 2; differences in sample collection site; sample handling (e.g., underfilled Li-heparin anticoagulant tubes, storage temperature prior to Li-heparin plasma separation, sample storage prior to analysis and reagents/assay methodology); and hemolysis [71,72,73,74]. Regardless, the higher GGT in Study 1 was considered an artifact and not the result of a physiological response to the test article in the rats.

## 5. Conclusions

In summary, BNT162b vaccine candidates were tolerated when administered IM to rats at 30 and/or 100 µg three times at weekly intervals. The nonclinical findings were comparable between the different vaccine candidates at the same dose level, and there was evidence of a dose effect on nonclinical findings and serology. The high reproducibility of observations made in two independent GLP toxicity studies using the same platform but different encoding antigens demonstrate that a platform toxicity approach would be suitable for the development of an mRNA-LNP vaccine. This conclusion is aligned with the recent WHO guidelines on the development of mRNA vaccines for infectious diseases [75,76,77] that suggests platform data may support entry into clinical trials with a different antigen as long as the platform has not significantly changed. Further, because the nonclinical findings were likely driven in part by the innate immune response to the RNA-LNP rather than the antigen, we suggest that a platform approach may be considered for characterized mRNA-LNP vaccines for entry into Phase I clinical trials. 

## Figures and Tables

**Figure 1 vaccines-11-00417-f001:**
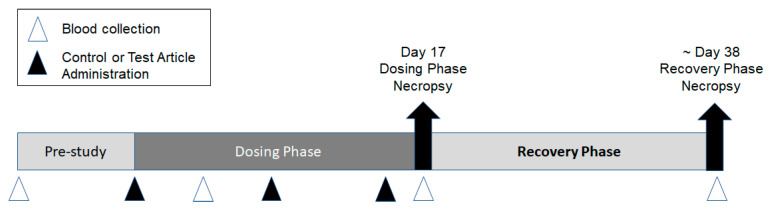
Overview of experimental design. Male and female Wistar Han rats (15/sex/group) were administered 3 once weekly intramuscular injections (
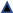
) of control (saline or vehicle), BNT162b1 (30 or 100 µg), BNT162b2 V8 (100 µg), BNT162b2 V9 (30 µg), or BNT162b3 (30 µg). Animals were euthanized two days after the last dose, Day 17 (10 animals/sex/group), and approximately 3 weeks later (Day 38). Blood (Δ) was collected for serology 
analysis prior to dose initiation and at each necropsy. Blood was collected for clinical pathology assessments on Day 3 and at each necropsy.

**Figure 2 vaccines-11-00417-f002:**
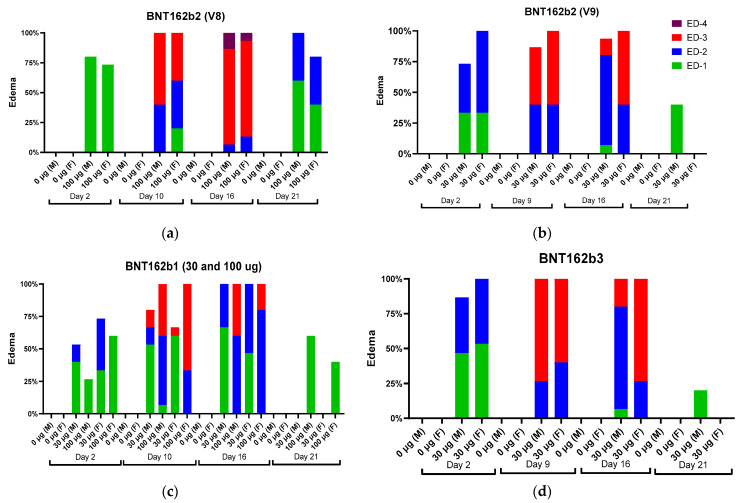
Edema incidence and severity in male (M) and female (F) Wistar Han rats immunized intramuscularly (IM) with (**a**) BNT162b2 V9 (30 µg), (**b**) BNT162b2 V8 (100 µg), (**c**) BNT162b1 (30 µg or 100 µg), and (**d**) BNT162b3 (30 µg), or buffer control. Edema measurements from Dosing Phase Days 2, 9, 16, and 21 for BNT162b2 V9 (30 µg) and BNT162b3 (30 µg). Edema measurements from Dosing Phase Days 2, 10, 16, and 21 for BNT162b2 V8 (100 µg) and BNT162b1 (30 µg or 100 µg).

**Figure 3 vaccines-11-00417-f003:**
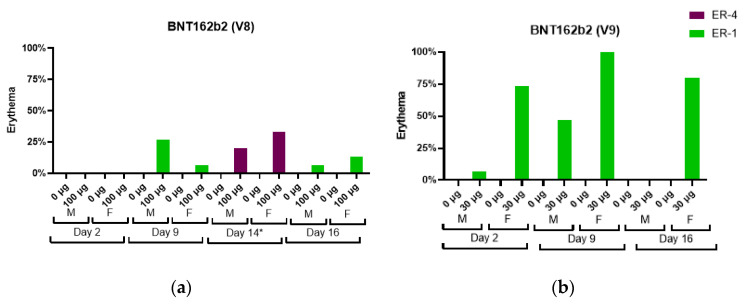
Erythema incidence and severity in male (M) and female (F) Wistar Han rats immunized intramuscularly (IM) with (**a**) BNT162b2 V8 (100 µg), (**b**) BNT162b2 V9 (30 µg), (**c**) BNT162b3 (30 µg), and (**d**) BNT162b1 (30 µg or 100 µg) or buffer control. Erythema measurements from Dosing Phase Days 2, 3/5 *, 9, 14 *, and 16. * Days 3/5 and 14 are only available for Study 1.

**Figure 4 vaccines-11-00417-f004:**
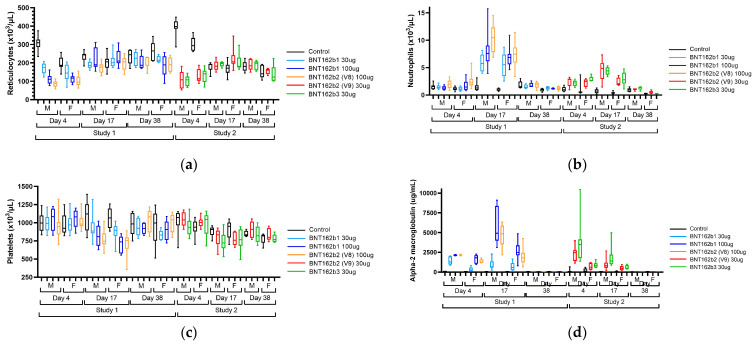
Reticulocyte counts (**a**), neutrophil counts (**b**), platelet counts (**c**), and alpha-2 macroglobulin (**d**) concentrations in male (M) and female (F) Wistar Han rats immunized intramuscularly (IM) with BNT162b1 (30 µg or 100 µg), BNT162b2 V8 (100 µg), BNT162b2 V9 (30 µg), BNT162b3 (30 µg), or buffer control. Blood samples were collected on Dosing Phase Days 4, 17, and 38. Box-whisker plots show group medians (middle line), 25th and 75th percentiles (box), and min and max (lower and upper whiskers).

**Figure 5 vaccines-11-00417-f005:**
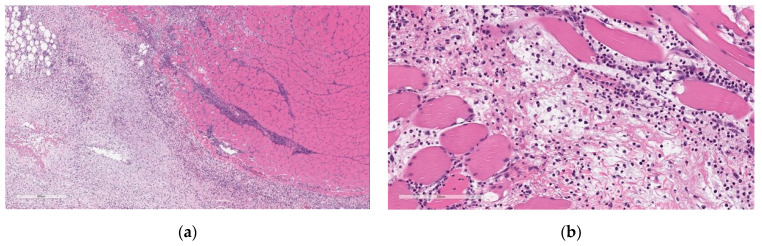
Histopathologic features of Injection site inflammation and edema at the end of the dosing phase. (**a**) Injection site sections of BNT162b2 (V9) treated animals euthanized 2 days after the third dose (Day 17) showed inflammatory cells admixed with abundant pale eosinophilic fluid (edema) infiltrating and expanding subcutaneous tissue and connective tissue of skeletal muscle. (**b**) Higher magnification of (**a**) showing.

**Figure 6 vaccines-11-00417-f006:**
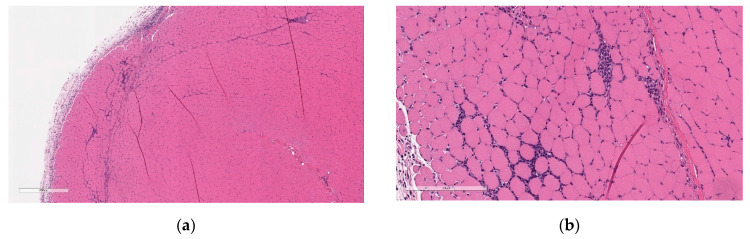
Histopathologic features of Injection site inflammation at the end of recovery. (**a**) Injection site sections of BNT162b2 (V9) treated animals euthanized after 3-week recovery showed fewer inflammatory cells infiltrating and expanding subcutaneous tissue and spaces around skeletal myofibers. (**b**) Higher magnification of (**a**) showing inflammatory cells (fewer plasma cells and lymphocytes) surrounding myofibers. Data are scanned from Study 2.

**Figure 7 vaccines-11-00417-f007:**
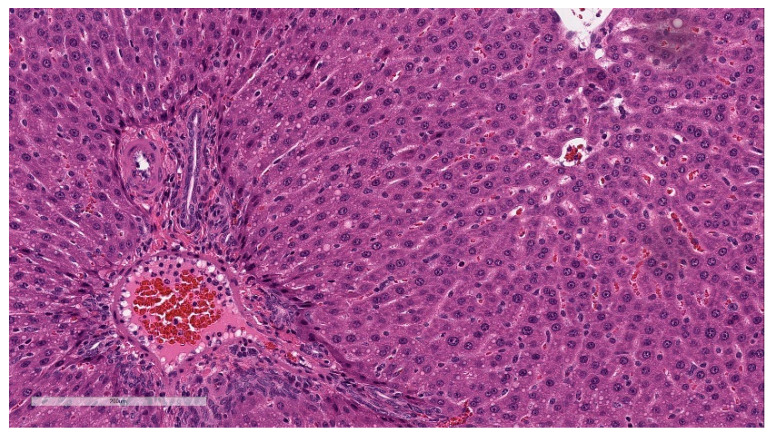
Histopathologic features of the liver at the end of the dosing phase. Liver sections of BNT162b2 (V9) treated animals euthanized 2 days after the third dose (Day 17) showed hepatocytes containing small clear round membrane-bound structures within the cytoplasm (inset). Data are scanned from Study 2.

**Figure 8 vaccines-11-00417-f008:**
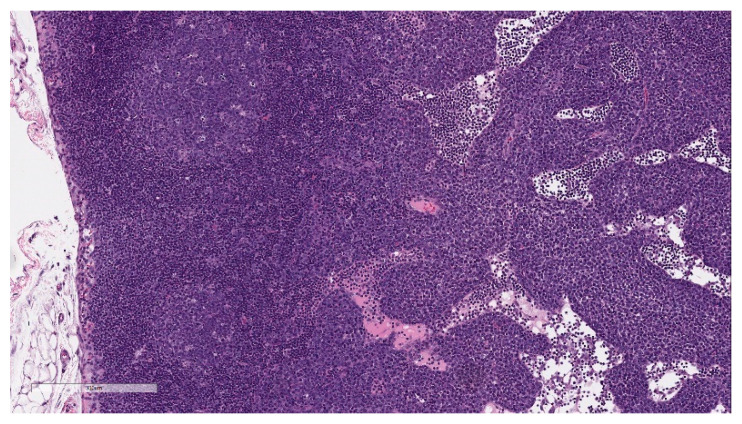
Histopathologic features of draining lymph node at the end of the dosing phase. Lymph node sections of BNT162b2 (V9) vaccinated animal showing prominent germinal centers (GC) and inflammatory cells (predominately plasma cells) expanding sinuses (inset). Data are scanned from Study 2.

**Figure 9 vaccines-11-00417-f009:**
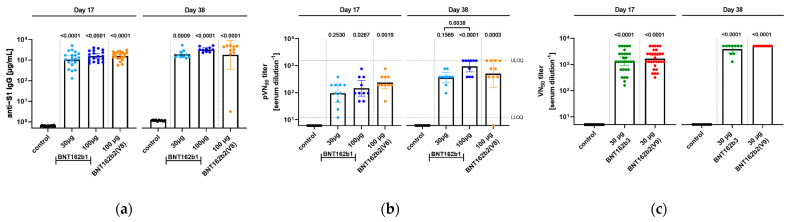
Wistar Han rats were immunized intramuscularly (IM) with BNT162b1, BNT162b2, or BNT162b3 vaccine candidates or buffer control. Analysis of serum samples was performed 17 days [*n* = 20] or 38 days [(**a**,**b**): *n* = 10; (**c**): *n* =5] after first immunization. Geometric means of each group ± 95% confidence interval (CI) are included, and *p*-values are given to control or determine if significance occurred as indicated (One-Way ANOVA of mixed-effect analysis using Tukey’s multiple comparisons test). (**a**), S1-specific IgG levels in sera of rats determined by ELISA. (**b**), 90% pseudovirus neutralization (pVN90) titers in sera of rats immunized with the different BNT162b vaccine candidates. (**c**), 50% SARS-CoV-2 neutralization (VN50) titer in sera of rats immunized with different BNT162b vaccine candidates.

**Table 1 vaccines-11-00417-t001:** Description of BNT162b modRNA vaccine candidates.

BNT162 Vaccine Candidate (Product Code)	SARS-CoV-2 Variant ^a^	Encoded Antigen	Sequence Variant ^b^
BNT162b1	Wild type	SARS-CoV-2 RBD, a secreted variant	V5
BNT162b2	Wild type	Full-length SARS-CoV-2 S protein bearing mutations preserving neutralization-sensitive sites	V8 and V9 ^c^
BNT162b3	Wild type	SARS-CoV-2 RBD, a membrane-bound variant	V5TM

^a^ Wild type refers to the USA-WA1/2020 strain. ^b^ Sequence variant refers to the nucleotide sequence of the RNA component encoding the antigen. ^c^ Note that there were two variants of the BNT162b2 vaccine tested. The RNA component of the two sequence variants (V8 and V9) have different nucleotide sequences, but both encode the same antigen. Abbreviations: RBD = receptor binding domain; S protein = spike protein; V = variant.

## Data Availability

The data presented in this study are available in the article.

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
