# Peer review of "Toxicological Assessments of a Pandemic COVID-19 Vaccine—Demonstrating the Suitability of a Platform Approach for mRNA Vaccines"

_vaccines, 2023, doi:10.3390/vaccines11020417_

Round 1

Reviewer 1 Report

In their manuscript entitled "Toxicological assessments of a pandemic COVID-19 vaccine – 2 demonstrating the suitability of a platform approach for mRNA 3 vaccines", the authors investigated the acute toxicity and the immunogenicity of mRNA vaccines jointly developed by Pfizer and BioNTech. This study showed that the mRNA based SARS-CoV-2 vaccines are well tolerated and of appreciable immunogenicity in a rat model. Generally, the study is well designed and nicely conducted.  I have two minor comment:

1, Please specify how the infiltrated cell type in muscle tissue is ascertained.

2, Considering that the BNT vaccine has been licensed for human use for two years, have these data been reported before?

Author Response

Response to Reviewer 1 Comments

In their manuscript entitled "Toxicological assessments of a pandemic COVID-19 vaccine – 2 demonstrating the suitability of a platform approach for mRNA 3 vaccines", the authors investigated the acute toxicity and the immunogenicity of mRNA vaccines jointly developed by Pfizer and BioNTech. This study showed that the mRNA based SARS-CoV-2 vaccines are well tolerated and of appreciable immunogenicity in a rat model. Generally, the study is well designed and nicely conducted. I have two minor comment:

Point 1. Please specify how the infiltrated cell type in muscle tissue is ascertained.

Response to Point 1: Thank you for this comment. All collected tissues were examined microscopically by a board-certified veterinary pathologist, who identified cell types by their morphology.  Cell type identification was based on the morphology of the cell nucleus, cell size, the appearance of the cytoplasm, and, in case of granulocytes, granule staining. We have added the following sentences in the material and method section in response to the reviewer’s comment, “The type of infiltrating cells in tissues was based on the morphology of their nucleus, their size, appearance of cytoplasm, and in case of granulocytes how their granules stain.” (Added text on page 7, lines 320-322.)

Point 2. Considering that the BNT vaccine has been licensed for human use for two years, have these data been reported before?

Response to Point 2: This is the first time that a detailed description of the repeat-dose toxicity study designs and results for BNT162b2 and several other Pfizer-BioNTech COVID-19 vaccine candidates using the modified-nucleoside RNA platform have been shared. A very brief, high-level description of the repeat-dose toxicity study designs and results for BNT162b2 only were mentioned in a summary of a scientific session “COVID-19 Therapeutics and Vaccines: A Race to Save Lives” from the 2021 Society of Toxicology Annual meeting, which was published in Toxicological Sciences (available at: https://academic.oup.com/toxsci/article/185/2/119/6420720?login=true) in Nov 2021. This same level of detail is publicly available in the European Medicines Agency EPAR Public Assessment Report for Comirnaty (available at: https://www.ema.europa.eu/en/documents/assessment-report/comirnaty-epar-public-assessment-report_en.pdf) and/or US FDA Summary Basis for Regulatory Action for Comirnaty (available at: https://www.fda.gov/media/151733/download). As our manuscript provides a much higher level of detail of the repeat-dose toxicity study results as well as covers the results from other BNT162b vaccine candidates supporting that a platform approach can be used with this type of vaccine, we believe this manuscript provides new and valuable data to the field of vaccine development.

Reviewer 2 Report

The authors of this manuscript report two studies to evaluate the safety and immunogenicity of four COVID-19 mRNA vaccine candidates in Wistar Han rats.

The results proved that four BNT162b mRNA vaccine candidates were tolerated in rats when administered IM once weekly (3 total doses) at doses of 30 and/or 100 μg, and induced dose-related immune responses.

Although the results of the clinical trial of the Pfizer-BioNTech COVID-19 vaccine had been published already, it is still interesting to virologists.

The studies were designed carefully, and the manuscript is well-written.

It would be easier to understand if the data of days 17 and 28 in Figure 6 can be integrated into one clustered column chart.

Reviewer 3 Report

In this paper, authors tested the mRNA vaccine candidates in Wistar Han rats and recorded the immune and inflammatory responses. The overall results indicate rats have tolerated the vaccine candidates safely and support the vaccine candidates to clinical development. In general, the manuscript is well written with enough information and logic. There are still a few things to improve.

1.     In your texts, for example, line 34-47, 52-60, 61-62, 65-67, 80-86, you don’t have enough references. Usually, we add citations after each solid statement. Please go over your entire manuscript and add citations accordingly. 

2.     Body temperature is a key sign after taking vaccines. Please apply your experimental results to human conditions and discuss more. 

3.     Even though your study mainly focused on host responses and if the mRNA vaccine candidates are suitable, please also discuss the vaccine efficiency in the paper. If you haven’t done any experiment showing the efficiency of the vaccine candidates preventing host from getting the viruses, please still discuss using any results that has been revealed already in the field. 

4.     Maybe I missed that, but I didn’t see anything about rat mortality/survival. The rats you used are probably healthy. However, in real human life, there are people with underlying health issues, and it is important to have experimental group rats to represent that too. Well, that’s something to consider. But still, please do consider and discuss about any potential risks.

Author Response

Response to Reviewer 3 Comments

In this paper, authors tested the mRNA vaccine candidates in Wistar Han rats and recorded the immune and inflammatory responses. The overall results indicate rats have tolerated the vaccine candidates safely and support the vaccine candidates to clinical development. In general, the manuscript is well written with enough information and logic. There are still a few things to improve.

Point 1. In your texts, for example, line 34-47, 52-60, 61-62, 65-67, 80-86, you don’t have enough references. Usually, we add citations after each solid statement. Please go over your entire manuscript and add citations accordingly.

Response to Point 1: Additional references have been added throughout the manuscript, especially in the Introduction and Discussion Sections.

Point 2. Body temperature is a key sign after taking vaccines. Please apply your experimental results to human conditions and discuss more.

Response to Point 2: As described in the manuscript, vaccine-dependent reversible body temperature elevation of approximately 1°C post immunization was observed. These observations indicated that a temperature elevation in humans post vaccination was possible, but that severe fever was not expected. A note on this topic as well as potential correlation to the Phase 3 clinical trial results of fever has been added to the discussion section.

Point 3. Even though your study mainly focused on host responses and if the mRNA vaccine candidates are suitable, please also discuss the vaccine efficiency in the paper. If you haven’t done any experiment showing the efficiency of the vaccine candidates preventing host from getting the viruses, please still discuss using any results that has been revealed already in the field.

Response to Point 3: The focus of this manuscript is the nonclinical safety evaluation of the COVID-19 modified-nucleoside RNA vaccine candidates. Antigen-specific responses are monitored as part of the repeat-dose toxicity study to ensure that there is a response in the animal model in which the toxicity is being evaluated. The efficacy of the response to BNT162b2 and at least one other COVID-19 modified-nucleoside RNA vaccine candidate is described in other publications: nonclinical efficacy by Vogel et al 2021 and clinical trial efficacy in Polack et al 2020. These papers are referenced in the current manuscript, and we believe it is beyond the scope of this manuscript to discuss this further as it is not the focus of the data being presented and vaccine efficacy has been discussed in the above referenced publications as well as in other peer-reviewed publications.

References:

Vogel AB; Kanevsky I; Che Y, et al. BNT162b vaccines protect rhesus macaques from SARS-CoV-2. Nature 2021; 592:283-89.

Polack FP; Thomas SJ; Kitchin N; et al. Safety and Efficacy of the BNT162b2 mRNA Covid-19 Vaccine. NEJM 2020;383:2603-15.

Point 4. Maybe I missed that, but I didn’t see anything about rat mortality/survival. The rats you used are probably healthy. However, in real human life, there are people with underlying health issues, and it is important to have experimental group rats to represent that too. Well, that’s something to consider. But still, please do consider and discuss about any potential risks.

Response to Point 4: There was no vaccine related mortality in the studies and the manuscript has been edited to address this.

Regulatory guidances for infectious disease vaccine development do not require nonclinical safety evaluation of vaccines in animal models of disease (WHO, 2005; WHO, 2014). Given the potential and known differences in immune systems between species, this type of analysis is best addressed in the clinic or via real-world evidence studies.

References:

World Health Organization. Annex 1. Guidelines on the nonclinical evaluation of vaccine. In: WHO Technical Report Series No. 927, Geneva, Switzerland; World Health Organization; 2005:31-63.

World Health Organization. Annex 2. Guidelines on the nonclinical evaluation of vaccine adjuvants and adjuvanted vaccines. In: WHO Technical Report Series No. 987. Geneva, Switzerland: World Health Organization; 2014:59-100. 2014.